# The association between circadian syndrome and possible sarcopenia in an aging population: A 4-year follow-up study

Qian Zhong[1,☯], Li Ren[2,☯], Tianhong Wang[1,3], Zhenmei An[1,*], Yusi Hua[4,*]

**1** Department of Endocrinology, West China Hospital, Sichuan University, Chengdu, China, **2** Division of Vascular Surgery, Department of General Surgery, West China Hospital, Sichuan University, Chengdu, China, **3** Department of Clinical Research, West China Hospital, Sichuan University, Chengdu, China, **4** Department of Anesthesiology, West China Hospital, Sichuan University, Chengdu, Sichuan Province, China

☯ These authors contributed equally to this work.
* azmhxnfm@163.com (ZA); yusihua@wchscu.cn (YH)

## Abstract

### Introduction

Recently, circadian syndrome (CircS) is proposed as a novel risk cluster based on sleep disorder, depression, dyslipidemia, hyperglycemia, hypertension and abdominal obesity. To investigate the association between CircS and possible sarcopenia, this study was performed.

### Methods

Possible sarcopenia is defined according to Asian Working Group for Sarcopenia in 2019, which includes measures of muscle strength and physical performance. In the baseline survey, 7,905 participants aged ≥ 40 years from the China Health and Retirement Longitudinal Study were included. Multivariate logistic regression was used to evaluate the association between CircS and possible sarcopenia. Subgroup and interactive analyses were adopted to verify the findings in the overall population and identify potential interactive effects. The obese population was excluded and the missing values were interpolated using multivariate imputation by chained equations as sensitivity analyses. In addition, the participants were followed up for four years to explore the longitudinal association between CircS and incident possible sarcopenia.

### Results

As per one increase of CircS components, participants had a 1.11-fold (95% CI = 1.07–1.14, $P < 0.001$) risk of prevalent possible sarcopenia in the full model. The CircS group was associated with a 1.30-fold (95% CI = 1.17–1.44) risk of prevalent possible sarcopenia ($P < 0.001$). No significant interactive effects of covariates on the

**Data availability statement:** All the data are available from the China Health and Retirement Longitudinal Study (http://charls.pku.edu.cn/). All the researchers would be able to access these data in the same manner as the authors and that the authors did not have any special access privileges that others would not have.

**Funding:** The author(s) received no specific funding for this work.

**Competing interests:** The authors have declared that no competing interests exist.

association between CircS and prevalent possible sarcopenia were detected (all *P* for interaction > 0.05). All the subgroup and sensitivity analyses supported the positive association between CircS and possible sarcopenia. In the longitudinal follow-up, the odd ratio was 1.06 (95% CI = 1.00–1.13, *P* < 0.05) as per one increase of CircS components in the full model. The CircS group was also found to have an elevated risk of incident possible sarcopenia (odd ratio = 1.24, 95% CI = 1.03–1.50, *P* < 0.05) after adjusting all the covariates.

## Conclusions

CircS is a risk factor for possible sarcopenia, which may serve as a predictor of possible sarcopenia for early identification and intervention.

---

## 1 Introduction

The concept of sarcopenia was first introduced by American physician Irwin Rosenberg in 1989 [1,2], indicating age-related loss of skeletal muscle mass. In 2018, the European Working Group on Sarcopenia in Older People (EWGSOP2) updated the definition of sarcopenia, considering it as a condition of muscle failure characterized by age-related declines in overall skeletal muscle mass, strength, and function [3]. As sarcopenia research has progressed, various pieces of evidence suggest a close association between sarcopenia and limitations in the physical function of older individuals, potentially leading to impaired physical activities, increased risks of falls, fractures, hospitalization, and even premature death [4,5]. In China, the overall prevalence of sarcopenia among community-dwelling elderly individuals aged 80 and above is 26.6%, with rates of 33.3% in males and 21.7% in females [6]. According to the criteria set by the Asian Working Group for Sarcopenia (AWGS), the estimated prevalence of sarcopenia in Asia ranges from 4.1% to 11.5% [7]. It is estimated that around 50 million people worldwide are affected by sarcopenia, and this number is expected to increase to as high as 500 million by the year 2050 [8]. With the advent of the aging population, sarcopenia has become a significant global public health concern.

However, in mainstream practice, sarcopenia often has not received adequate attention. The AWGS 2019 criteria emphasized the importance of developing methods for early identification of those at risk of sarcopenia in community settings without the need for advanced diagnostic tests [9]. This update also introduced the term "possible sarcopenia," defined as low muscle strength with or without a decline in physical performance. Given the potential reversibility of this condition, early detection of older adults at risk of possible sarcopenia can facilitate timely health education, lifestyle interventions, and referral for confirmatory diagnosis [10].

In higher photosensitive organisms, the circadian system is a functional system composed of photosensitive neurons, the endocrine system, and clock genes that regulate circadian oscillations, enabling organisms to establish day-night rhythms on different levels, spanning from gene expression to behavior [11]. "Circadian syndrome

(CircS)" refers to the concept that living organisms, across various levels of complexity—from molecules and cells to entire organisms and populations—have evolved adaptive mechanisms. These mechanisms result in periodic fluctuations in biological activities in response to the daily changes in environmental conditions. Building on this concept, CircS has been proposed as a novel risk cluster, characterized by factors such as reduced sleep duration, abdominal obesity, depression, hypertension, dyslipidemia, and hyperglycemia [12]. The circadian system consists of two main components: the central clock and peripheral clocks [13]. The central clock is controlled by the suprachiasmatic nucleus (SCN) located in the hypothalamus, which is highly sensitive to light [14]. It generates a circadian rhythm with a 24-hour cycle, influencing various physiological activities in animals, including eating patterns and digestion, physical activity, body temperature, blood pressure, cellular regeneration, hormone secretion, sleep-wake cycles, immune function and more [15]. Peripheral clocks exist in various tissues and cells outside the SCN and are capable of regulating their own internal rhythms [16]. For example, peripheral organs such as skeletal muscles [17], liver [18] and kidney [19] have their own peripheral clocks. In the case of skeletal muscles, their rhythms are not only controlled by the SCN but also influenced by factors such as meal timing, regular physical exercise, and sleep patterns [20]. This intrinsic self-regulation plays a crucial role in various aspects of skeletal muscle, including growth, strength, and function [21]. Importantly, disruptions in these peripheral clocks are closely associated with muscle-related conditions, such as sarcopenia and muscle atrophy [22].

Currently, there is increasing evidence that modern lifestyles, such as sleep deprivation, lack of physical activity, high-energy diets, and shift work, as well as the use of artificial lighting, may lead to disruptions and disturbances in circadian rhythms, resulting in different health outcomes [23]. These factors are believed to be important contributors to the development of sarcopenia. Recently, a systematic meta-analysis concluded that sleep quality can predict the risk of developing sarcopenia [24]. A later sleep timing, which can result in CircS as well as adversely affect sleep quality, has been demonstrated to be linked with sarcopenia in middle-aged individuals [25]. Furthermore, a prospective cohort study in South Korea found a significantly increased risk of sarcopenia among shift workers [26]. However, there has been limited research considering potential determinants of sarcopenia, especially modifiable factors, which are crucial for developing targeted interventions to mitigate sarcopenia in older adults with CircS.

Therefore, based on the updated definition of sarcopenia proposed by the AWGS 2019, we conducted cross-sectional and longitudinal analyses using nationally representative data from the China Health and Retirement Longitudinal Study (CHARLS) in 2011 and 2015 to investigate the risk of "possible sarcopenia" among Chinese older adults with circadian syndrome.

## 2 Materials and methods

### 2.1 Study population

This study derived from CHARLS 2011 baseline survey and 2015 follow-up survey. CHARLS is an ongoing survey targeting the aging population in China. At the baseline survey in 2011, the participants were selected using a multistage probability sampling method. Finally, 17,705 participants were sampled from 28 provinces covering 150 counties and 400 villages. Information regarding the medical histories, physical examination, blood biomarkers, economic status, health care and insurance, etc. were collected by well-trained medical staffs. The participants were followed up every two or three years in 2013, 2015 and 2018. However, only the 2011 and 2015 waves had blood biomarkers and the 2018 wave did not perform physical examination. Thus, the status of CircS and sarcopenia can only be evaluated in 2011 and 2015, which was used to construct the follow-up cohort in this study. Detailed information of CHARLS with regard to its study design, sampling method, summary of collected data etc. can be obtained in one previous study [27] or official website (http://charls.pku.edu.cn/). CHARLS was approved by the ethical review board of Peking University (IRB 00001052–11014). Written and oral informed consent was obtained from all participants before their participation.

The process of data cleansing was summarized in Fig 1. Among the 17,705 participants in 2011, 13 participants were first excluded due to lack information of gender. Then, 124 participants aged < 40 years or without information of age were

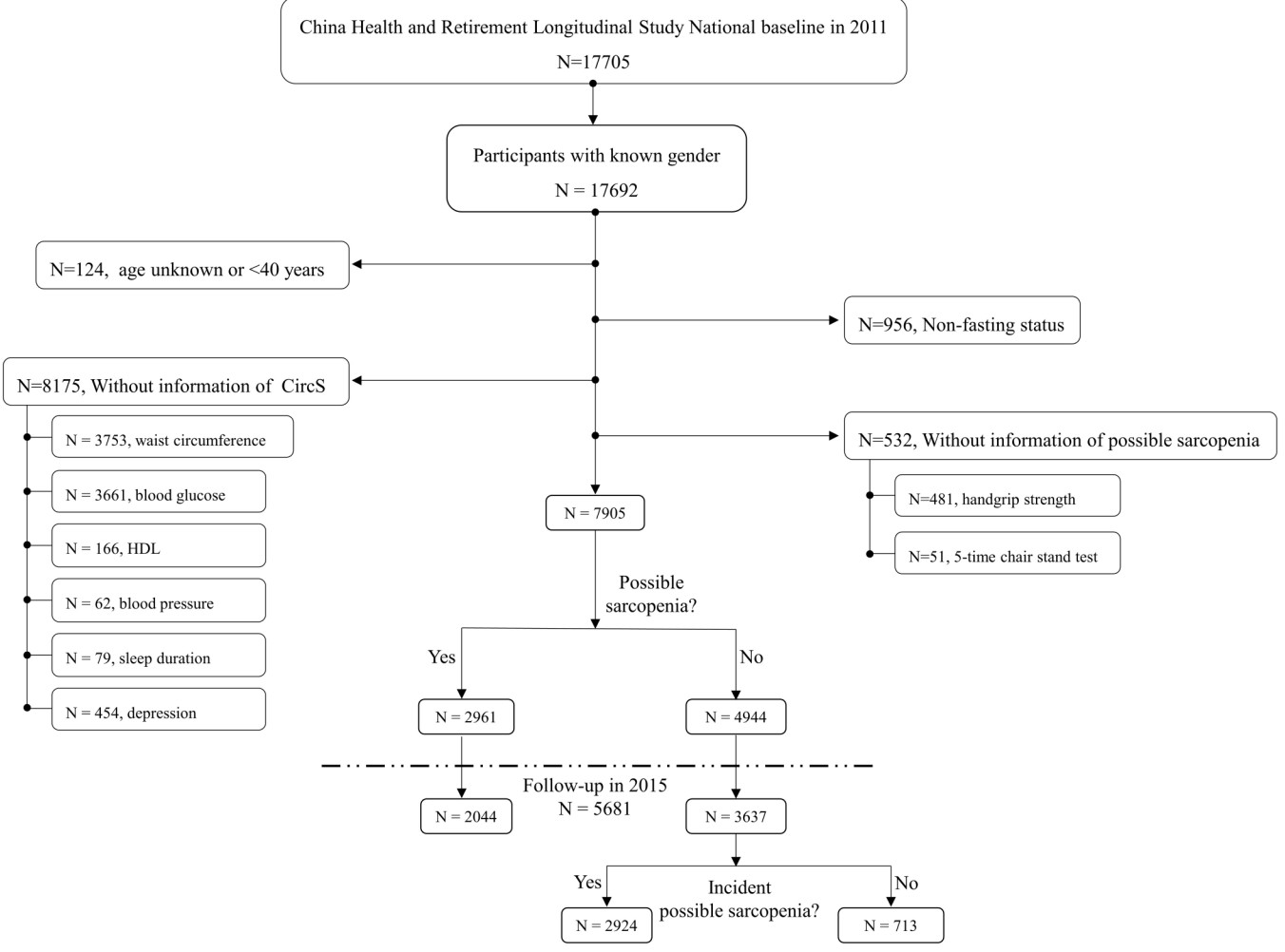

**Fig 1. Study design and analysis strategy.** The criteria and detailed numbers of excluded participants are showed in this figure. After data cleansing, 7905 participants were included to investigate the cross-sectional association between CircS and prevalent sarcopenia in 2011. The longitudinal association was explored in a four-year follow-up survey from 2011-2015.

also deleted. To evaluate the blood biomarkers, 956 participants were excluded due to be non-fasting status when drawing venous blood. In addition, 8,175 and 532 participants were removed due to without information of CircS and possible sarcopenia, respectively. Finally, 7,905 respondents were included in the final cross-sectional analysis and 5681 of them were followed up for four years till 2015.

## 2.2 Measurements of CircS

The term, CircS, is built based on seven components: central obesity, elevated triglycerides and glucose, decreased HDL, hypertension, reduced sleep duration and depression. Central obesity was defined as waist circumference ≥ 85 cm for males or ≥ 80 cm for females [28]. To diagnose CircS, blood biomarkers including triglyceride, glucose and HDL were determined. Participants were asked to be fasted overnight. A 6-mL tube was used to collect venous blood, which was then centrifuged to separate plasma. Oxidase method, hexokinase and direct method were used to determine the levels of triglyceride, glucose and HDL, respectively [29]. The cut-off for diagnosing elevated triglycerides or glucose were

set at 150 mg dL$^{-1}$ and 100 mg dL$^{-1}$. Meanwhile, participants receiving medication for triglycerides and glucose elevation can also be diagnosed with elevated triglycerides or glucose regardless of their levels. Decreased HDL was defined as HDL < 40 mg dL$^{-1}$ for males or < 50 mg dL$^{-1}$ for females or drug treatment for reduced HDL. The pressure was assessed using HEM-7200 electronic monitor (Omron, Dalian, China) by experienced nurses. Each participant was assessed for three times and the average was calculated as the final pressure for diagnose. Hypertension was defined as systolic pressure ≥ 130 mmHg or diastolic pressure ≥ 85 mmHg or drug treatment of hypertension. Participants with self-reported sleep duration < 6 hours were diagnosed as reduced sleep duration. To evaluate the depressive status, a widely used epidemiological questionnaire, Center for Epidemiological Studies Depression Scale-10 (CESD-10) [30], was used. Respondents with score ≥ 10 can be defined as having depression. Participants can be diagnosed as CircS who have at least four components concurrently [23].

## 2.3 Assessment of possible sarcopenia

The possible sarcopenic status was assessed according to the AWGS 2019 [9], which includes measures of muscle strength and physical performance. Participants with low muscle strength or low physical performance were defined as possible sarcopenia [9].

The muscle strength was indicated using handgrip strength (Kg) as recommended by AWGS. Handgrip strength was assessed using a TM WL-1000 dynamometer (Nantong Yuejian Physical Measurement Instrument Co., Ltd., Nantong, China). In a standing position, participants tried their best to squeeze the dynamometer for a few seconds using dominant and non-dominant hands in turn. Each hand was measured twice. The maximum handgrip strengths of the left and right hands were extracted and then averaged as the final handgrip strength of the participants. As recommended by AWGS, the criteria for defining low handgrip strength were < 28 Kg for males and < 18 Kg for females.

To assess physical performance, 5-time chair stand test (CST) was adopted. The participants were asked to keep their arms folded across their chest from a chair and then rise and sit down continuously five times. The time was recorded in a unit of second. According to the standard from AWGS, participants with CST ≥ 12 s were defined as low physical performance. In this study, participants who tried the 5-time CST but failed were also diagnosed as low physical performance as one previous study did [31].

## 2.4 Assessment of covariates

Covariates including demographic variables, health-related behaviors and some blood biomarkers were adjusted in this study. Demographic variables included age (years), gender (male or female), marital status (married/cohabitating or widowed/separated/divorced) and educational levels (literate or illiterate). The widowed/separated/divorced were clustered as others group. Respondents with educational levels higher than elementary school were clustered as literate. Health-related behaviors included cigarette consumption (current, never or ex-smoker), alcohol consumption (more than once a month, less than once a month, or never) and afternoon nap (yes or no). In the survey, participants were asked "During the past month, how long did you take a nap after lunch in general?". Those who answered with a "0" were defined as "No" and those with afternoon nap regardless of the duration were defined as "Yes". Blood biomarkers included low-density lipoprotein (mg/dL), total cholesterol (mg/dL), uric acid (mg/dL), C-reactive protein (mg/L) and blood creatinine (mg/dL).

## 2.5 Statistical analyses

Participants were divided into normal group or CircS group. Baseline characteristics across groups were presented as mean ± standard error (SD) for continuous variables with normal distribution and median (25–75% quantiles) with non-normal distribution, and n (%) for categorical variables. Student-t, Wilcoxon or Chi-square tests were used to test the differences between the normal group and CircS group according to data types.

Given that exposure (CircS) and outcome (possible sarcopenia) are all binary variables, multivariable binary logistic regression was used to investigate the cross-sectional and longitudinal associations between CircS and possible sarcopenia. Four models were built by adjusting for different covariates. Model 1 was a crude model. Model 2 was adjusted for demographic variables. Model 3 was further adjusted for health-related behaviors. Model 4 was further adjusted for blood biomarkers. To control the discrepancies of covariates across groups, we also used propensity scores matching (PSM) and inverse probability of treatment weighting (IPTW) methods. In PSM, the nearest neighbor matching with a caliper of 0.03 was employed. In IPTW, confounders were weighted by the inverse of propensity scores. The absolute standardized bias was calculated to evaluate the effectiveness of PSM and IPTW.

To verify the robustness of the findings, some sensitivity analyses were performed. First, the number of CircS components was used as a continuous variable to examine the incremental relationship between CircS components and risk of possible sarcopenia. Second, subgroup and interactive analyses were performed to investigate potential interactive effects of covariates. Third, we also excluded the obese population (body mass index > 28.0 $Kg/m^2$) to verify the association in the non-obese population. Finally, given the missing values in some covariates (#para10437_32718232103), the dataset was interpolated using the multivariate imputation by chained equations based on random forest methods [30]. All analyses in this study were performed using STATA software (Version 16.1, Stata Corporation) and R studio (Version 4.0.2). The significant threshold was defined as $P$-value < 0.05 (two-sided).

## 3 Results

### 3.1 Characteristics of participants in the baseline survey

A total of 7,905 participants were included in the cross-sectional analysis (Fig 1). Among them, 3,064 respondents (38.64%) had CircS, who tended to be older, female, divorced/widowed/divorced or divorced (all $P$ < 0.001, Table 1). In addition, the CircS group tended to be obese and have less cigarette and alcohol consumption ($P$ < 0.001). As for blood biomarkers, the CircS had significantly higher uric acid, low-density lipoprotein, total cholesterol and C-reactive protein levels than the normal group ($P$ < 0.01). However, there was no difference of blood creatinine between the two groups ($P$ = 0.107).

### 3.2 The cross-sectional association between CircS and prevalent possible sarcopenia

As summarized in Table 2, the CircS group had a significantly higher risk of prevalent possible sarcopenia. In the crude model, it was found that participants had a 1.15-fold (95% CI = 1.12–1.18, $P$ < 0.001) risk of prevalent possible sarcopenia as per one increase of CircS components. In the full model (model 4), the odds ratio (OR) slightly declined to 1.11 (95% CI = 1.07–1.14) but still remained significant ($P$ < 0.001). As a categorical variable, the ORs were 1.44 (95% CI = 1.31–1.58), 1.23 (95% CI = 1.11–1.36), 1.22 (95% CI = 1.11–1.35) and 1.30 (95% CI = 1.17–1.44) for model 1, model 2, model 3 and model 4, respectively (all $P$ < 0.001).

To balance the inter-group differences of covariates, PSM and IPTW were used. As shown in S2 Fig and S3 Fig, the absolute standardized biases were all close to null, indicating well-balanced covariates. The biases from confounding factors were restricted. After PSM, it was revealed that the CircS group had a 1.24-fold (95% CI = 1.11–1.39, $P$ < 0.001) risk of possible sarcopenia and the OR for the CircS group was 1.27 (95% CI = 1.19–1.36, $P$ < 0.001) in IPTW.

### 3.3 Association between CircS and prevalent possible sarcopenia in subgroup and interactive analyses

Fig 2 depicted the results of subgroup and interactive analyses. A significantly increased risk of possible sarcopenia was observed in all the subgroups of age (< 60 years and ≥ 60 years), gender (male and female), marital status (married/cohabitating and others), educational levels (literate and illiterate), cigarette consumption (current smoker, non-smoker and ex-smoker) and afternoon nap (yes and no). However, in alcohol consumption, the ORs were 1.21 (95%

**Table 1. Characteristics of participants in the baseline survey.**

| Characteristics | Normal N = 4861 | CircS N = 3064 | Overall N = 7905 | P |
|---|---|---|---|---|
| Age (years) | 58.49±9.50 | 60.04±9.28 | 59.09±9.45 | <0.001 |
| Gender | | | | |
| Male | 2592 (53.54%) | 1089 (35.54%) | 3681 (46.57%) | <0.001 |
| Female | 2249 (46.46%) | 1975 (64.46%) | 4224 (53.43%) | |
| Marital status | | | | |
| Married/cohabitating | 4151 (85.75%) | 2496 (81.46%) | 6647 (84.09%) | <0.001 |
| Others | 690 (14.25%) | 568 (18.54%) | 1258 (15.91%) | |
| Educational levels | | | | |
| Literate | 2735 (56.50%) | 1546 (50.46%) | 4281 (54.16%) | <0.001 |
| Illiterate | 2106 (43.50%) | 1518 (49.54%) | 3624 (45.84%) | |
| BMI (Kg/m$^2$) | | | | |
| <18.5 | 429 (8.90%) | 76 (2.50%) | 505 (6.43%) | <0.001 |
| 18.5-24.0 | 3000 (62.25%) | 1055 (34.73%) | 4055 (51.61%) | |
| 24.0-28.0 | 1086 (22.54%) | 1259 (41.44%) | 2345 (29.85%) | |
| ≥28.0 | 304 (6.31%) | 648 (21.33%) | 952 (12.12%) | |
| Cigarette consumption | | | | |
| Current smoker | 1708 (35.29%) | 687 (22.42%) | 2395 (30.30%) | <0.001 |
| Non-smoker | 2712 (56.03%) | 2087 (68.11%) | 4799 (60.72%) | |
| Ex-smoker | 420 (8.68%) | 290 (9.46%) | 710 (8.98%) | |
| Alcohol consumption | | | | |
| Drink more than once a month | 1397 (28.86%) | 558 (18.21%) | 1955 (24.73%) | <0.001 |
| Drink but less than once a month | 418 (8.63%) | 227 (7.41%) | 645 (8.16%) | |
| None of these | 3026 (62.51%) | 2279 (74.38%) | 5305 (67.11%) | |
| Afternoon nap | | | | |
| No | 2235 (46.17%) | 1359 (44.37%) | 3594 (45.47%) | 0.117 |
| Yes | 2606 (53.83%) | 1704 (55.63%) | 4310 (54.53%) | |
| Uric acid (mg/dL) | 4.36±1.22 | 4.58±1.30 | 4.44±1.26 | <0.001 |
| Creatinine (mg/dL) | 0.78±0.20 | 0.78±0.21 | 0.78±0.20 | 0.107 |
| LDL cholesterol (mg/dL) | 116.50±32.43 | 119.12±38.96 | 117.51±35.09 | 0.001 |
| Total cholesterol (mg/dL) | 189.99±36.24 | 200.97±42.40 | 194.21±39.09 | <0.001 |
| C-reactive protein (mg/L) | 0.86 (0.49-1.78) | 1.34 (0.71-2.69) | 1.04 (0.55-2.14) | <0.001 |
| Depression | 1330 (27.47%) | 1613 (52.64%) | 2943 (37.23%) | <0.001 |
| Reduced sleep duration | 961 (19.85%) | 1336 (43.60%) | 2297 (29.06%) | <0.001 |
| Abdominal obesity | 2015 (41.62%) | 2634 (85.97%) | 4649 (58.81%) | <0.001 |
| Systolic pressure (mmHg) | 125.93±22.81 | 138.53±26.35 | 130.79±25.00 | <0.001 |
| Diastolic pressure (mmHg) | 73.46±11.41 | 79.34±11.95 | 75.72±11.97 | <0.001 |
| Hypertension | 1786 (36.89%) | 2310 (75.39%) | 4096 (51.82%) | <0.001 |
| Elevated blood glucose | 2107 (43.52%) | 2503 (81.69%) | 4610 (58.32%) | <0.001 |
| Reduced HDL | 1006 (20.78%) | 2293 (74.84%) | 3299 (41.73%) | <0.001 |
| Elevated triglyceride | 445 (9.19%) | 1881 (61.39%) | 2326 (29.42%) | <0.001 |
| Handgrip strength (Kg) | 32.34±9.83 | 29.75±10.09 | 31.34±10.01 | <0.001 |
| 5-time chair stand test (seconds) | 10.36±3.96 | 11.26±4.42 | 10.71±4.16 | <0.001 |
| Possible sarcopenia | 1653 (34.15%) | 1308 (42.69%) | 2961 (37.46%) | <0.001 |

BMI: body mass index; LDL: low-density lipoprotein; HDL: high density lipoprotein.

**Table 2. The cross-sectional association between CircS and prevalent sarcopenia.**

| Models | Per one of CircS component (continous) | | CircS (yes versus no) | |
| --- | --- | --- | --- | --- |
| | OR (95% CI) | P | OR (95% CI) | P |
| Model 1 | 1.15 (1.12-1.18) | <0.001 | 1.44 (1.31-1.58) | <0.001 |
| Model 2 | 1.08 (1.05-1.12) | <0.001 | 1.23 (1.11-1.36) | <0.001 |
| Model 3 | 1.08 (1.05-1.12) | <0.001 | 1.22 (1.11-1.35) | <0.001 |
| Model 4 | 1.11 (1.07-1.14) | <0.001 | 1.30 (1.17-1.44) | <0.001 |
| PSM | – | – | 1.24 (1.11-1.39) | <0.001 |
| IPTW | – | – | 1.27 (1.19-1.36) | <0.001 |

Model 1: crude model; Model 2: adjusting for age, gender, marital status and educational levels; Model 3: further adjusting for cigarette and alcohol consumption and afternoon nap; Model 4: adjusting for blood biomarkers including LDL, total cholesterol, blood uric acid, CRP and blood creatinine. PSM: propensity scores matching. PSM: propensity scores matching; IPTW: inverse probability of treatment weighting.

CI = 0.96–1.52, $P = 0.115$) and 1.45 (95% CI = 0.97–2.15, $P = 0.067$) for participants drinking more than once a month and less than once a month, respectively. The elevated and insignificant OR may be attributed the relatively limited sample size in the two subgroups. In addition, no significant interactive effects of covariates on the association between CircS and prevalent possible sarcopenia were detected (all $P$ for interaction > 0.05).

### 3.4 Sensitivity analyses

To further verify the findings in this study, some sensitivity analyses were performed. First, the obese population (body mass index > 28.0 Kg/m$^2$) were excluded (Fig 3A). In the full model, participants had a 1.10-fold (95% CI = 1.06–1.14, $P < 0.001$) risk of prevalent possible sarcopenia as per one increase of CircS components. As a categorical variable, suffering CircS was associated with a higher risk of prevalent possible sarcopenia (OR = 1.27, 95% CI = 1.13–1.42, $P < 0.001$) after adjusting all the covariates.

In addition, there were missing values ranging from 0% to 0.7% in the covariates (S1 Fig). Although the proportion of missing values is very small and less likelihood to bias the results, we still interpolated the datasets using random forest to validate the findings (Fig 3B). As per one increase of CircS components, the ORs were 1.15 (95% CI = 1.12–1.18), 1.08 (95% CI = 1.05–1.12), 1.08 (95% CI = 1.05–1.12) and 1.11 (95% CI = 1.07–1.14) in the crude model, model 2, model 3 and the full model (all $P < 0.001$). As a categorical variable, a 1.29-fold risk of possible sarcopenia (95% CI = 1.16–1.43, $P < 0.001$) was identified for the CircS group.

### 3.5 The longitudinal association between CircS and incident possible sarcopenia

To investigate the longitudinal association between CircS and incident possible sarcopenia, the participants were followed up for four years till 2015. In Table 3, as per one increase of CircS components, the ORs were 1.10 (95% CI = 1.05–1.16, $P < 0.001$), 1.05 (95% CI = 1.00–1.11, $P = 0.073$), 1.05 (95% CI = 0.99–1.11, $P = 0.095$) and 1.06 (95% CI = 1.00–1.13, $P < 0.05$) in the crude model, model 2, model 3 and the full model. As a categorical variable, the CircS group was found to have an elevated risk of incident possible sarcopenia (OR = 1.24, 95% CI = 1.03–1.50, $P < 0.05$) in the full model.

## 4 Discussion

So far, there have been few studies investigating the relationship between CircS and possible sarcopenia in the middle-aged and elderly population. To the best of our knowledge, this study represents the first nationwide cross-sectional investigation and a 4-year longitudinal observation of the Chinese elderly population, aimed at confirming CircS as a potential risk factor for possible sarcopenia.

| Subgroups | Normal group (as reference) | CircS group OR (95%CI) | | P | P for interaction |
|---|---|---|---|---|---|
| **Age group** | | | | | 0.638 |
| <60 | 1.00 | 1.34 (1.16-1.54) | | <0.001 | |
| ≥60 | 1.00 | 1.31 (1.12-1.53) | | <0.001 | |
| **Gender** | | | | | 0.247 |
| Male | 1.00 | 1.41 (1.20-1.66) | | <0.001 | |
| Female | 1.00 | 1.21 (1.06-1.39) | | 0.006 | |
| **Marital status** | | | | | 0.113 |
| Married/cohabitating | 1.00 | 1.25 (1.11-1.40) | | <0.001 | |
| Others | 1.00 | 1.58 (1.22-2.03) | | <0.001 | |
| **Educational levels** | | | | | 0.478 |
| Literate | 1.00 | 1.36 (1.17-1.58) | | <0.001 | |
| Illiterate | 1.00 | 1.24 (1.07-1.44) | | 0.004 | |
| **Cigarette consumption** | | | | | 0.112 |
| Current smoker | 1.00 | 1.42 (1.16-1.74) | | <0.001 | |
| Non-smoker | 1.00 | 1.20 (1.05-1.37) | | 0.006 | |
| Ex-smoker | 1.00 | 1.73 (1.21-2.48) | | 0.003 | |
| **Alcohol consumption** | | | | | 0.676 |
| Drink more than once a month | 1.00 | 1.21 (0.96-1.52) | | 0.115 | |
| Drink but less than once a month | 1.00 | 1.45 (0.97-2.15) | | 0.067 | |
| None of these | 1.00 | 1.31 (1.16-1.48) | | <0.001 | |
| **Afternoon nap** | | | | | 0.474 |
| No | 1.00 | 1.24 (1.06-1.44) | | 0.008 | |
| Yes | 1.00 | 1.35 (1.17-1.55) | | <0.001 | |

1.0  1.5  2.0  2.5

**Fig 2. Association between CircS and prevalent sarcopenia in subgroup and interactive analyses.** In the multivariable logistic regression models, covariates were adjusted as model 4 in previous analyses except for subgroup variables.

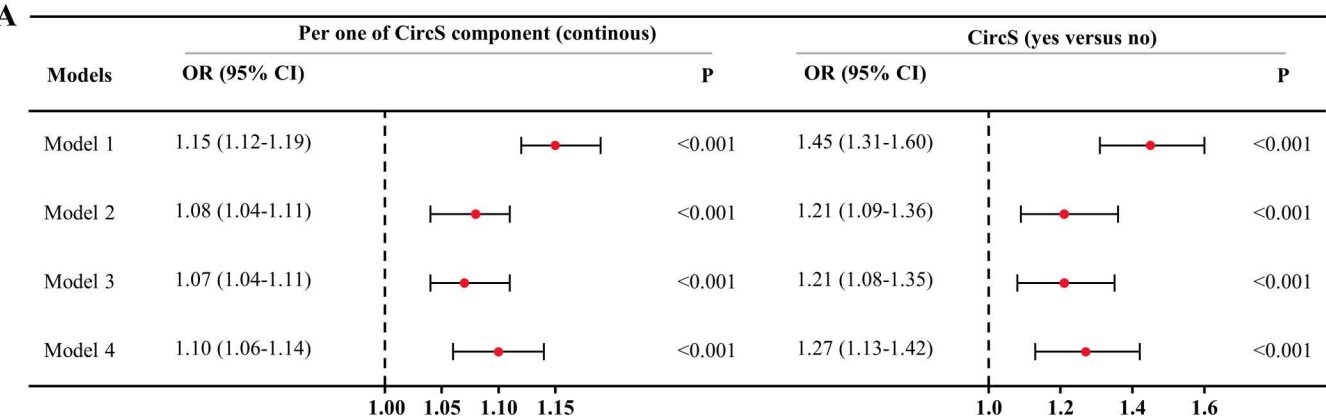

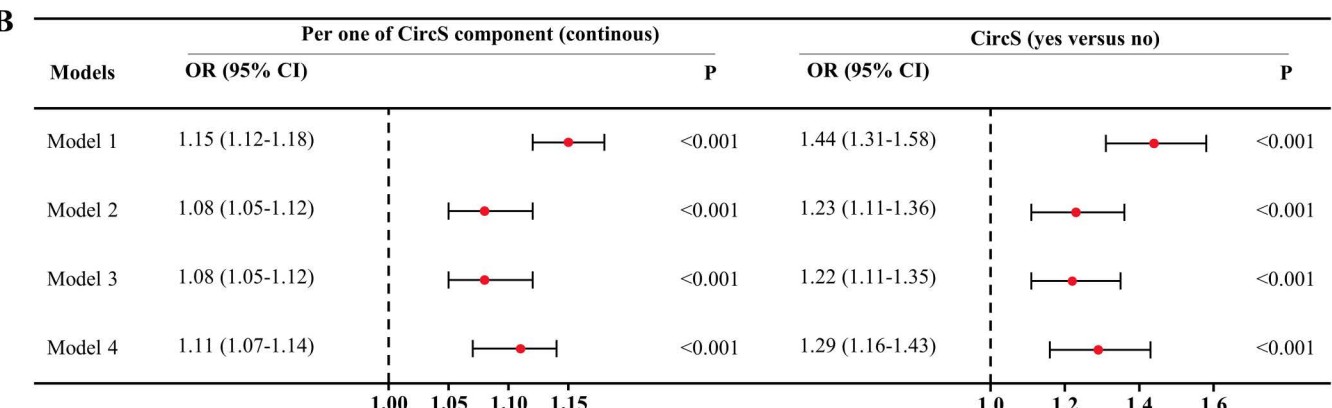

**Fig 3. Sensitivity analyses excluding the obese population and interpolating the missing values.** A depicts the results of multivariate logistic regression excluding the obese population (body mass index ≥ 28.0 Kg/m²). B shows the results of multivariate logistic regression after interpolating the missing values by random forest. Model 1: crude model; Model 2: adjusting for age, gender, marital status and educational levels; Model 3: further adjusting for cigarette and alcohol consumption and afternoon nap; Model 4: adjusting for blood biomarkers including low-density lipoprotein, total cholesterol, blood uric acid, C-reactive protein and blood creatinine.

**Table 3. The longitudinal association between CircS and incident sarcopenia.**

| Models | Per one of CircS component (continous) | | CircS (yes versus no) | |
|---|---|---|---|---|
| | OR (95% CI) | P | OR (95% CI) | P |
| Model 1 | 1.10 (1.05-1.16) | <0.001 | 1.34 (1.13-1.59) | 0.001 |
| Model 2 | 1.05 (1.00-1.11) | 0.073 | 1.21 (1.01-1.45) | 0.036 |
| Model 3 | 1.05 (0.99-1.11) | 0.095 | 1.20 (1.01-1.44) | 0.043 |
| Model 4 | 1.06 (1.00-1.13) | 0.040 | 1.24 (1.03-1.50) | 0.022 |

Model 1: crude model; Model 2: adjusting for age, gender, marital status and educational levels; Model 3: further adjusting for cigarette and alcohol consumption and afternoon nap; Model 4: adjusting for blood biomarkers including LDL, total cholesterol, blood uric acid, CRP and blood creatinine.

In this nationally representative cohort study, we discovered that CircS may increase the risk of possible sarcopenia, even after adjusting for potential confounding factors. Our baseline survey results indicated that individuals in the CircS group tended to be older, female, divorced/separated/widowed, obese, less likely to smoke and drink alcohol, but had

higher LDL, total cholesterol, CRP, and uric acid levels. They also had a higher prevalence of depression, reduced sleep duration, hypertension, and elevated blood glucose. It can be speculated that older women experiencing menopause, low sleep quality, and depression are at a higher risk of possible sarcopenia.

Previous research on CircS and sarcopenia has predominantly focused on specific behaviors, including irregular or short-duration sleep, shift work, and exposure to artificial lighting [32]. There have been fewer studies that employ the CircS concept to categorize individuals and explore its association with health outcomes, despite its examination from an endocrinological perspective in molecular studies [26]. When examining the relationship between sleep duration and sarcopenia, research has uncovered a potential link between sleep-related issues (duration, quality, and timing) and sarcopenia [33]. Recent population-based and laboratory studies have revealed a U-shaped relationship between sleep duration and the prevalence of sarcopenia [34–38]. The recommended daily sleep duration for healthy adults is 7–9 hours [39], and both insufficient and excessive sleep duration can disrupt the natural circadian rhythm, leading to phase shifts. Hu et al. [35] found that compared to the normal sleep duration group (6–8 hours), the prevalence of sarcopenia increased to 27.5% in the short sleep group (< 6 hours) and 22.2% in the long sleep group (> 8 hours), with a more pronounced effect observed in females ($P = 0.014$) and no statistically significant gender difference in males ($P = 0.356$). Moreover, research conducted by several scholars has confirmed that chronic later sleep times and poor sleep quality serve as risk factors for sarcopenia [25]. Buchmann and colleagues [37], using the Pittsburgh Sleep Quality Index (PSQI) to assess sleep quality, grip strength to evaluate muscle strength, and DXA to assess appendicular lean mass (ALM), found that low sleep quality was associated with decreased grip strength ($P = 0.009$) and low ALM ($P = 0.016$). Lucassen et al. [25] further supported the association between reduced sleep quality and sarcopenia. Their research indicated that for each one-unit increase in PSQI score, the risk of developing sarcopenia increased by 10%. Regarding shift work, research has demonstrated that individuals employed in shift work face an elevated risk of circadian rhythm disruption [40], and shift workers are 1.7 times more likely to develop sarcopenia compared to those who do not work in shifts [26]. A recent study found that CircS is associated with physical frailty in middle-aged and older adults living in the community, with sarcopenia being a major component of physical frailty [41].

Moreover, CircS is considered a significant underlying cause of Metabolic Syndrome (MetS) and should be recognized alongside the MetS cluster and its comorbidities, including sleep disorders and depression [42]. There has been a growing body of research exploring the association between sarcopenia and MetS, as well as its components. A 2012 study from Japan reported a positive correlation between MetS and sarcopenia in men aged 65–74 (OR = 5.5, OR 95%CI: 1.9–15.9) [43]. Other studies have found that low muscle mass is associated with MetS in both young men and women, regardless of obesity status [44–46]. Several studies focused on the elderly population have shown a negative correlation between MetS and muscle mass [47,48]. Regarding muscle strength, research indicates a negative association between MetS and muscle strength in young men and women [49,50]. Notably, our previous research has demonstrated that depression can lead to sarcopenia [51]. Sleep quality and duration are also significantly linked to the development of sarcopenia, suggesting that restoring the body's circadian rhythm could be a crucial approach to improving skeletal muscle function and treating sarcopenia. These associations suggest that the inclusion of these factors in CircS may enhance its predictive value for possible sarcopenia compared to MetS alone. However, whether CircS is a better predictor of potential sarcopenia than MetS requires further investigation in future studies. Several pathophysiological mechanisms can elucidate the relationship between CircS and the increased risk of possible sarcopenia. In addition to being controlled by the central circadian clock, skeletal muscles are regulated by peripheral clocks, which establish their own rhythms [52]. These skeletal muscle self-rhythms are driven by clock genes within muscle cells, and their expression is in a dynamic equilibrium. Disruption of one of these genes, whether due to deficiency or overexpression, may lead to disruptions in muscle growth, development, functionality, and metabolism [53]. In skeletal muscle, the core clock genes that play a major role include *Bmal1* [54], *Clock* [55], and *Per* [56]. Research indicated that mice with global (whole-body) *Bmal1* knockout (KO) exhibited decreased muscle weight, body weight, and lifespan with disrupted circadian rhythms [57] and severe sarcopenia at

40 weeks of age, primarily attributed to reduced muscle fiber diameter and alterations in fiber type [54]. Studies have also demonstrated that mice with *Clock* gene defects exhibit decreased muscle strength and endurance due to reduced mitochondrial content and impaired mitochondrial function [58,59]. Additionally, Bae et al. [60] indicated that m*Per2* knockout mice manifest muscle-related disorders, such as reduced locomotor endurance, contractile dysfunction, and movement impairments.

Myogenic regulatory factors (MRFs) play a crucial role in the proliferation, growth, and differentiation of myoblasts into mature skeletal muscle fibers. MRFs include MyoD and MyoG [5]. Previous research has confirmed that the expression of MyoD is directly controlled by the clock genes *Baml1* and *Clock*. The core enhancer in the MyoD promoter is recognized by the heterodimer of BAML1 and CLOCK, which activates transcription, promoting myoblast differentiation into skeletal muscle fibers [20]. Moreover, the expression of MyoG exhibits periodic characteristics similar to MyoD [61]. In skeletal muscle, the *Bmal1* gene is a key positive regulatory factor that promotes myogenic differentiation by controlling MRFs [62]. It is through this control of MRFs that the circadian rhythm indirectly participates in the regulation of skeletal muscle growth and development. Furthermore, recent evidence suggests a crosstalk between the circadian clock and skeletal muscle metabolism [63]. A preliminary study indicates that the circadian clock, MyoD, and metabolic factors (such as *PGC-1*) provide a potential feedback loop system that could play a crucial role in both the maintenance and adaptation of skeletal muscle [64].

Furthermore, the reduction in both sleep duration and quality, alterations in circadian rhythm, and increased frequency of sleep disorders can lead to decreased secretion of growth hormone (GH), insulin-like growth factor-1 (IGF-1), and testosterone [65–67]. These changes can also trigger a low-level inflammatory response in the body [68]. Intermittent hypoxia can further induce a chronic stress response and stimulate the nocturnal secretion of cortisol [69,70]. All of these factors are associated with the development of sarcopenia [25]. Additionally, diminished sleep duration and quality impact the regulation of insulin secretion and sensitivity, thereby increasing the risk of sarcopenia [71].

The primary strength of this study lies in the use of a large-scale nationally representative longitudinal database. However, the study also has several limitations. Firstly, information on sleep duration was obtained through retrospective self-reports rather than objective measurements of sleep parameters, which may introduce recall bias. Prior research has noted differences between self-report and actigraphy measurements of sleep duration, though consistency in older adults remains relatively high, ranging from 66% to 78%. Secondly, due to the lack of specific questions related to the SARC-F questionnaire in CHARLS, we did not employ the SARC-F or SARC-CalF questionnaires for screening potential sarcopenia. Instead, we directly assessed muscle strength and physical performance to detect possible sarcopenia. Future research endeavors may explore the development and validation of SARC-F using analogous questions collected through the CHARLS dataset. Additionally, since our study focused solely on possible sarcopenia, we did not include muscle mass measurements. Future research should explore this aspect to provide a more comprehensive assessment of sarcopenic status. Thirdly, participants who did not undergo baseline physical function tests may be frailer and more susceptible to possible sarcopenia. Consequently, this study might underestimate the prevalence of possible sarcopenia. However, since participants with possible sarcopenia at baseline were excluded from the analysis, the estimation of the incidence of possible sarcopenia is unlikely to be significantly affected. Fourthly, the exclusion of some participants due to missing data from CST, grip strength, and/or sleep time could introduce selection bias into the results. Fifth, in the course of a 4-year longitudinal observation of an elderly population, the attrition in the panel over time is unavoidably not completely random. Losing track of participants during follow-up might also introduce sample selection bias. In addition, despite adjusting for numerous potential confounding factors, the possibility of residual confounding cannot be entirely eliminated. Sixthly, the follow-up period was relatively short, and a 4-year follow-up may not be sufficient to detect more pronounced effects of CircS. Finally, although current evidence suggests a significant association between CircS and the development of sarcopenia, the underlying mechanisms remain unclear and require further investigation.

## 5 Conclusion

In summary, our study offers strong evidence of the association between CircS and an elevated risk of possible sarcopenia. Therefore, early assessment of sarcopenia risk in patients with CircS, along with individualized interventions targeting influencing factors is crucial for preventing the onset and progression of sarcopenia and improving patient quality of life. This warrants significant attention in clinical practice.

## Supporting information

**S1 Fig. The missing values of covariates.** A shows the percentages of missing values of covariates. B shows the combinations of missing values of covariates.
(JPG)

**S2 Fig. Absolute standardized bias across covariates before and after PSM in the baseline survey.** Absolute standardized bias < 10% is accepted as a well-balanced dataset between covariates. In this figure, all the biases were close to null after PSM.
(JPG)

**S3 Fig. Absolute standardized bias across covariates before and after IPTW in the baseline survey.** Absolute standardized bias < 10% is accepted as a well-balanced dataset between covariates. In this figure, all the biases were close to null after IPTW.
(JPG)

## Author contributions

**Conceptualization:** Yusi Hua, Qian Zhong, Li Ren, Zhenmei An.

**Data curation:** Qian Zhong, Li Ren, Tianhong Wang.

**Formal analysis:** Qian Zhong, Li Ren.

**Supervision:** Yusi Hua, Zhenmei An.

**Writing – original draft:** Qian Zhong, Li Ren.

**Writing – review & editing:** Yusi Hua, Zhenmei An.

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
