## [Decision Letter · Decision Letter 0]

11 Feb 2025

PONE-D-25-00708The association between circadian syndrome and possible sarcopenia in an aging population: A 4-year follow-up studyPLOS ONE

Dear Dr. hua,

Thank you for submitting your manuscript to PLOS ONE. After careful consideration, we feel that it has merit but does not fully meet PLOS ONE’s publication criteria as it currently stands. Therefore, we invite you to submit a revised version of the manuscript that addresses the points raised during the review process.

We look forward to receiving your revised manuscript.

Kind regards,

Emiliano Cè, Ph.D.

Academic Editor

PLOS ONE

Journal Requirements:

https://www.frontiersin.org/journals/medicine/articles/10.3389/fmed.2021.755705/full

https://www.mdpi.com/1422-0067/21/9/3106

https://www.mdpi.com/2077-0383/12/22/7073

In your revision ensure you cite all your sources (including your own works), and quote or rephrase any duplicated text outside the methods section. Further consideration is dependent on these concerns being addressed.

Additional Editor Comments:

Dear Authors,

two experts in the field reviewed your manuscript reporting several major issues you should consider during the revision process.

Reviewers' comments:

Reviewer's Responses to Questions

**Comments to the Author**

1. Is the manuscript technically sound, and do the data support the conclusions?

Reviewer #1: Yes

Reviewer #2: Yes

2. Has the statistical analysis been performed appropriately and rigorously? 

Reviewer #1: Yes

Reviewer #2: Yes

3. Have the authors made all data underlying the findings in their manuscript fully available?

Reviewer #1: Yes

Reviewer #2: Yes

4. Is the manuscript presented in an intelligible fashion and written in standard English?

Reviewer #1: Yes

Reviewer #2: Yes

5. Review Comments to the Author

Reviewer #1: This study investigated the association between circadian syndrome (CircS) and possible sarcopenia in older Chinese adults using data from the China Health and Retirement Longitudinal Study. The researchers found that CircS was associated with an increased risk of prevalent and incident possible sarcopenia in cross-sectional and longitudinal analyses. The findings suggest CircS may be a risk factor for possible sarcopenia in older adults and highlight the importance of evaluating sarcopenic conditions in elderly populations with CircS. My specific comments are listed below:

1. While the association between sleep disturbances and sarcopenia has been previously studied, this paper examines the broader concept of circadian syndrome in relation to possible sarcopenia, which is a relatively novel approach. However, the study does not provide substantial new mechanistic insights beyond what is already known about circadian disruption and muscle health.

2. The use of a large, nationally representative longitudinal dataset is a key strength. However, the reliance on self-reported sleep duration rather than objective sleep measures is a significant limitation that may introduce recall bias and reduce the accuracy of CircS classification.

3. The study uses a composite definition of CircS that includes various metabolic and behavioral factors. While this aligns with previous literature, it may conflate the effects of circadian disruption with those of metabolic syndrome, making it difficult to isolate the specific impact of circadian rhythm disturbances on sarcopenia risk.

4. The use of handgrip strength and chair stand tests to assess possible sarcopenia is appropriate and aligns with AWGS guidelines. However, the lack of muscle mass measurements limits the ability to fully characterize sarcopenic status according to more comprehensive definitions.

5. The authors employed multiple statistical approaches, including propensity score matching and inverse probability weighting, which strengthens the robustness of their findings. However, the potential for residual confounding remains, particularly given the complex interplay between circadian rhythms, metabolic health, and muscle function.

6. The 4-year follow-up period provides valuable insight into the temporal relationship between CircS and sarcopenia development. However, the relatively short duration may not fully capture the long-term effects of circadian disruption on muscle health, which could develop over decades. Longer follow-up with multiple measurements are more appropriate to answer the research question.

7. While the study suggests that CircS may be a risk factor for possible sarcopenia, it does not provide clear guidance on how this information should be applied in clinical practice. The authors could have expanded on potential interventions or screening strategies that could be implemented based on their findings.

Reviewer #2: The authors have submitted a manuscript detailing the association between circadian syndrome and possible sarcopenia. While the title initially seems novel, I have significant concerns regarding how the authors have proposed this idea.

Q1: In line 81, Circadian Syndrome (CircS) is proposed as a novel risk cluster based on... Please add a reference.

Q2: Additionally, I believe “circadian system” and “circadian syndrome” refer to two different concepts. The title of the paper is “The Association between Circadian Syndrome and Possible Sarcopenia.” Please provide a more specific introduction to “Circadian Syndrome” rather than to the “circadian system.”

Q3: In line 188, what is the definition of an afternoon nap?

Q4: Medications can influence triglyceride and glucose levels, as well as HDL and blood pressure. What about the history of medication use?

Q5: Since circadian syndrome is similar to metabolic syndrome, does it better predict possible sarcopenia compared to metabolic syndrome? I believe this information would be valuable for clinical practice.

Q6: There are two components—short sleep and depression symptoms—that differentiate circadian syndrome from metabolic syndrome. How does the association between these two individual components and possible sarcopenia, as discussed by the author in line 341?

Q7: There is excessive discussion on the pathophysiological mechanisms of possible sarcopenia and circadian syndrome. It would be better to focus on comparing this study's findings with previous literature.

6. PLOS authors have the option to publish the peer review history of their article (what does this mean? ). If published, this will include your full peer review and any attached files.

**Do you want your identity to be public for this peer review?** For information about this choice, including consent withdrawal, please see our Privacy Policy .

Reviewer #1: No

Reviewer #2: No

---

## [Author Response · Author response to Decision Letter 0]

17 Mar 2025

Reviewer 1:

This study investigated the association between circadian syndrome (CircS) and possible sarcopenia in older Chinese adults using data from the China Health and Retirement Longitudinal Study. The researchers found that CircS was associated with an increased risk of prevalent and incident possible sarcopenia in cross-sectional and longitudinal analyses. The findings suggest CircS may be a risk factor for possible sarcopenia in older adults and highlight the importance of evaluating sarcopenic conditions in elderly populations with CircS. My specific comments are listed below:

1. While the association between sleep disturbances and sarcopenia has been previously studied, this paper examines the broader concept of circadian syndrome in relation to possible sarcopenia, which is a relatively novel approach. However, the study does not provide substantial new mechanistic insights beyond what is already known about circadian disruption and muscle health.

Response: Thank you for your feedback. We have addressed this point in the limitations section of the discussion to acknowledge this aspect of our study.

2. The use of a large, nationally representative longitudinal dataset is a key strength. However, the reliance on self-reported sleep duration rather than objective sleep measures is a significant limitation that may introduce recall bias and reduce the accuracy of CircS classification.

Response: Thank you for highlighting this point. We agree that while the use of a large, nationally representative longitudinal dataset is a key strength of our study, the reliance on self-reported sleep duration is a limitation that could introduce recall bias and affect the accuracy of CircS classification. We have addressed this limitation in the limitations section of the discussion.

3. The study uses a composite definition of CircS that includes various metabolic and behavioral factors. While this aligns with previous literature, it may conflate the effects of circadian disruption with those of metabolic syndrome, making it difficult to isolate the specific impact of circadian rhythm disturbances on sarcopenia risk.

Response: Thank you for your insightful comment. We have added a new section discussing MetS to better differentiate and compare CircS with MetS.

Revise: “Moreover, CircS is considered a significant underlying cause of Metabolic Syndrome (MetS) and should be recognized alongside the MetS cluster and its comorbidities, including sleep disorders and depression. There has been a growing body of research exploring the association between sarcopenia and MetS, as well as its components. A 2012 study from Japan reported a positive correlation between MetS and sarcopenia in men aged 65-74 (OR=5.5, OR 95%CI: 1.9-15.9). Other studies have found that low muscle mass is associated with MetS in both young men and women, regardless of obesity status. Several studies focused on the elderly population have shown a negative correlation between MetS and muscle mass. Regarding muscle strength, research indicates a negative association between MetS and muscle strength in young men and women. Notably, our previous research has demonstrated that depression can lead to sarcopenia. Sleep quality and duration are also significantly linked to the development of sarcopenia, suggesting that restoring the body’s circadian rhythm could be a crucial approach to improving skeletal muscle function and treating sarcopenia. These associations suggest that the inclusion of these factors in CircS may enhance its predictive value for possible sarcopenia compared to MetS alone. However, whether CircS is a better predictor of potential sarcopenia than MetS requires further investigation in future studies.”

4. The use of handgrip strength and chair stand tests to assess possible sarcopenia is appropriate and aligns with AWGS guidelines. However, the lack of muscle mass measurements limits the ability to fully characterize sarcopenic status according to more comprehensive definitions.

Response: Thank you for your feedback. We have addressed this point in the limitations section of the discussion to acknowledge this aspect of our study.

5. The authors employed multiple statistical approaches, including propensity score matching and inverse probability weighting, which strengthens the robustness of their findings. However, the potential for residual confounding remains, particularly given the complex interplay between circadian rhythms, metabolic health, and muscle function.

Response: Thank you for your feedback. We have acknowledged this point in the limitations section of the discussion to address this aspect of our study.

6. The 4-year follow-up period provides valuable insight into the temporal relationship between CircS and sarcopenia development. However, the relatively short duration may not fully capture the long-term effects of circadian disruption on muscle health, which could develop over decades. Longer follow-up with multiple measurements are more appropriate to answer the research question.

Response: Thank you for your feedback. We have acknowledged this point in the limitations section of the discussion to address this aspect of our study.

7. While the study suggests that CircS may be a risk factor for possible sarcopenia, it does not provide clear guidance on how this information should be applied in clinical practice. The authors could have expanded on potential interventions or screening strategies that could be implemented based on their findings.

Response: Thank you for your suggestion. We have supplemented the conclusion section to address this point.

Reviewer 2:

The authors have submitted a manuscript detailing the association between circadian syndrome and possible sarcopenia. While the title initially seems novel, I have significant concerns regarding how the authors have proposed this idea.

Q1: In line 81, Circadian Syndrome (CircS) is proposed as a novel risk cluster based on... Please add a reference.

Response: Thank you for your careful review of our manuscript. We have added the relevant reference at the specified location.

Q2: Additionally, I believe “circadian system” and “circadian syndrome” refer to two different concepts. The title of the paper is “The Association between Circadian Syndrome and Possible Sarcopenia.” Please provide a more specific introduction to “Circadian Syndrome” rather than to the “circadian system.”

Response: Thank you for your insightful comments. We have revised the introduction to provide a more specific and detailed explanation of “Circadian Syndrome”.

Revise: “Circadian syndrome (CircS)” refers to the concept that living organisms, across various levels of complexity—from molecules and cells to entire organisms and populations—have evolved adaptive mechanisms. These mechanisms result in periodic fluctuations in biological activities in response to the daily changes in environmental conditions. Building on this concept, CircS has been proposed as a novel risk cluster, characterized by factors such as reduced sleep duration, abdominal obesity, depression, hypertension, dyslipidemia, and hyperglycemia.

Q3: In line 188, what is the definition of an afternoon nap?

Response: Thanks for your kind notice. In the survey, participants were asked “During the past month, how long did you take a nap after lunch in general?”. Those who answered with a “0” were defined as “No afternoon nap” and those with afternoon nap regardless of the duration were defined as “Yes”. We have added relevant description in the manuscript.

Q4: Medications can influence triglyceride and glucose levels, as well as HDL and blood pressure. What about the history of medication use?

Response: This is a very critical concern. We apologize for our unclear statement. To mitigate the bias from the history of medication use, we also defined those with medication use for elevated triglyceride, glucose, and blood pressure, and decreased HDL, as patients. This is in line with the description in our first-draft of manuscript. In addition, this definition is also in line with the definition of Metabolic Syndrome

Q5: Since circadian syndrome is similar to metabolic syndrome, does it better predict possible sarcopenia compared to metabolic syndrome? I believe this information would be valuable for clinical practice.

Response: Thank you for your insightful question. While CircS shares similarities with MetS, it encompasses additional factors such as sleep disturbances and depression, which are also closely linked to muscle health. Although CircS may offer a broader perspective in predicting possible sarcopenia, further research is needed to determine whether it is a more accurate predictor than MetS. We have added this discussion to the manuscript.

Revise: “Moreover, CircS is considered a significant underlying cause of Metabolic Syndrome (MetS) and should be recognized alongside the MetS cluster and its comorbidities, including sleep disorders and depression. There has been a growing body of research exploring the association between sarcopenia and MetS, as well as its components. A 2012 study from Japan reported a positive correlation between MetS and sarcopenia in men aged 65-74 (OR=5.5, OR 95%CI: 1.9-15.9). Other studies have found that low muscle mass is associated with MetS in both young men and women, regardless of obesity status. Several studies focused on the elderly population have shown a negative correlation between MetS and muscle mass. Regarding muscle strength, research indicates a negative association between MetS and muscle strength in young men and women. Notably, our previous research has demonstrated that depression can lead to sarcopenia. Sleep quality and duration are also significantly linked to the development of sarcopenia, suggesting that restoring the body’s circadian rhythm could be a crucial approach to improving skeletal muscle function and treating sarcopenia. These associations suggest that the inclusion of these factors in CircS may enhance its predictive value for possible sarcopenia compared to MetS alone. However, whether CircS is a better predictor of potential sarcopenia than MetS requires further investigation in future studies.”

Q6: There are two components—short sleep and depression symptoms—that differentiate circadian syndrome from metabolic syndrome. How does the association between these two individual components and possible sarcopenia, as discussed by the author in line 341?

Response: Thank you for your question. We have incorporated this discussion into the manuscript.

Revise: “Notably, our previous research has demonstrated that depression can lead to sarcopenia. Sleep quality and duration are also significantly linked to the development of sarcopenia, suggesting that restoring the body’s circadian rhythm could be a crucial approach to improving skeletal muscle function and treating sarcopenia.”

Q7: There is excessive discussion on the pathophysiological mechanisms of possible sarcopenia and circadian syndrome. It would be better to focus on comparing this study's findings with previous literature.

Response: Thank you for your valuable feedback. We acknowledge that the discussion on the pathophysiological mechanisms of possible sarcopenia and CircS may be extensive. We have revised the discussion to emphasize a comparison of our study’s findings with previous literature and have removed some of the discussion on mechanisms.

---

## [Decision Letter · Decision Letter 1]

3 Apr 2025

The association between circadian syndrome and possible sarcopenia in an aging population: A 4-year follow-up study

PONE-D-25-00708R1

Dear Dr. hua,

We’re pleased to inform you that your manuscript has been judged scientifically suitable for publication and will be formally accepted for publication once it meets all outstanding technical requirements.

Kind regards,

Emiliano Cè, Ph.D.

Academic Editor

PLOS ONE

Additional Editor Comments (optional):

Reviewers' comments:

Reviewer's Responses to Questions

**Comments to the Author**

1. If the authors have adequately addressed your comments raised in a previous round of review and you feel that this manuscript is now acceptable for publication, you may indicate that here to bypass the “Comments to the Author” section, enter your conflict of interest statement in the “Confidential to Editor” section, and submit your "Accept" recommendation.

Reviewer #1: All comments have been addressed

Reviewer #2: All comments have been addressed

2. Is the manuscript technically sound, and do the data support the conclusions?

Reviewer #1: Yes

Reviewer #2: Yes

3. Has the statistical analysis been performed appropriately and rigorously? 

Reviewer #1: Yes

Reviewer #2: Yes

4. Have the authors made all data underlying the findings in their manuscript fully available?

Reviewer #1: Yes

Reviewer #2: Yes

5. Is the manuscript presented in an intelligible fashion and written in standard English?

Reviewer #1: Yes

Reviewer #2: Yes

6. Review Comments to the Author

Reviewer #1: The author has thoroughly addressed all the comments and suggestions provided in the previous round of review. After carefully going through the revised manuscript and the point-by-point responses, I am satisfied with the revisions made. I have no additional comments or concerns at this time and do not recommend any further changes.

Reviewer #2: The authors have addredded all comments I raised. This paper can be accepted now. Congrats to the authors.

7. PLOS authors have the option to publish the peer review history of their article (what does this mean? ). If published, this will include your full peer review and any attached files.

**Do you want your identity to be public for this peer review?** For information about this choice, including consent withdrawal, please see our Privacy Policy .

Reviewer #1: No

Reviewer #2: No

---

## [Editor Report · Acceptance letter]

PONE-D-25-00708R1

PLOS ONE

Dear Dr. hua,

I'm pleased to inform you that your manuscript has been deemed suitable for publication in PLOS ONE. Congratulations! Your manuscript is now being handed over to our production team.

Kind regards,

on behalf of

Prof. Emiliano Cè

Academic Editor

PLOS ONE